# Dietary Patterns of Patients with Prediabetes and Type 2 Diabetes

**DOI:** 10.3390/metabo13040532

**Published:** 2023-04-07

**Authors:** Camelia Oana Iatcu, Ana-Maria Gal, Mihai Covasa

**Affiliations:** 1College of Medicine, “Grigore T. Popa” University of Medicine and Pharmacy, 700115 Iasi, Romania; 2College of Medicine and Biological Sciences, Stefan cel Mare University of Suceava, 720229 Suceava, Romania; 3Department of Basic Medical Sciences, College of Osteopathic Medicine, Western University of Health Sciences, Pomona, CA 91766, USA

**Keywords:** dietary patterns, type 2 diabetes, prediabetes, factor analysis, EPIC-Norfolk FFQ

## Abstract

Given the worldwide high prevalence of type 2 diabetes, the prevention and control of this disease has become an urgent priority. In this research, we report the results from a cross-sectional study conducted in the counties of Suceava and Iasi, northeast of Romania, on 587 patients with type 2 diabetes and 264 patients with prediabetes. By employing a factor analysis (principal component) on 14 food groups followed by varimax orthogonal rotation, three dietary patterns were identified for each group. In prediabetes, a low adherence to a specific dietary pattern (1 and 2) was associated with lower fasting plasma glucose, blood pressure and serum insulin, compared to increased adherence. In patients with diabetes, a low adherence to Pattern 1 was associated with lower systolic blood pressures, while a low adherence to Pattern 3 was associated with a lower HbA1c, compared to high adherence. Statistically significant differences between the groups were observed for fats and oils, fish and fish products, fruit, potatoes, sugars, preserves and snacks intake. The study demonstrated that certain food patterns are associated with increased blood pressure, fasting blood glucose and serum insulin.

## 1. Introduction

Diabetes mellitus is a chronic condition that is characterized by chronic hyperglycemia, and occurs as a result of interactions between genetic and environmental factors [1]. Prediabetes is a high-risk state that is defined by fasting glucose levels of 100–125 mg/dL, glycated hemoglobin (HbA1c) of 5.7–6.4% or a glucose tolerance of 140–199 mg/dL two hours after ingesting a standardized 75 g glucose solution [2,3]. Risk factors for diabetes are both unmodifiable, such as genetics, age and race, as well as modifiable such as an unhealthy lifestyle and a lack of physical activity. Unhealthy eating, which includes excess caloric intake, is becoming more common, and is a characteristic of urbanization and economic growth [4]. Lifestyle changes are a cause of type 2 diabetes (T2DM) that promotes obesity, and the rapid increase in the number of obesity cases is the leading cause for the rapid growth of T2DM [5]. Currently, there are approximately 463 million patients with diabetes worldwide between 20 and 79 years of age [6], and diabetes is a major cause of death in people under the age of 60 [7]. It is estimated that this number could increase to 642 million by 2040 [6]. In Romania, the number of diabetic patients is increasing, and Romania is among the top ten countries in Europe in diabetes prevalence [8]. Given the importance of T2DM, including the high economic costs for its management, the prevention and control of this disease has become a worldwide priority [7]. Among the risk factors for diabetes, nutrition is the most important. Research has demonstrated significant associations between various foods and the risk of T2DM; however, such associations are complex, since people do not readily eat individual foods, but mixed foods instead. Therefore, the identification and analysis of dietary patterns has been frequently used to determine nutritional risk factors associated with diabetes, as well as to improve blood glucose management. Each dietary pattern represents a group of food items selected by individuals, and is influenced by culture, habits, tradition, restaurant settings, grocery availability and lifestyle, to name a few [9]. In particular, several studies have demonstrated the beneficial effects of the Mediterranean diet in patients with T2DM through its antioxidant and anti-inflammatory properties. As such, the Mediterranean diet contributes to the prevention of various diseases or complications, being known for its cardioprotective, neuroprotective, anticancer and antidiabetic roles [10,11].

To the best of our knowledge, no other study has been conducted in Romania to evaluate dietary patterns of prediabetic and diabetic patients, and to compare the dietary intake of patients with prediabetes with that of patients with T2DM. Therefore, the objectives of the present study were to evaluate dietary patterns and intake in prediabetic and T2DM patients, and to examine the effects of adherence to a specific dietary pattern on biochemical and clinical outcomes.

## 2. Subjects and Methods

### 2.1. Study Population

This cross-sectional study was conducted in 2020 in Suceava and Iasi counties, northeast of Romania. The study population consisted of 587 patients with T2DM and 264 patients with prediabetes. The participants were invited to attend a health examination at the Regional Hospitals of Suceava and Iasi, Diabetology clinic, where they were interviewed face-to-face by a trained interviewer using validated written questionnaires. The patients provided informed consent to participate in the study, and the study was approved by the University of Suceava Research Ethics Committee (protocol number 11733/14.07.2020), and the University of Medicine and Pharmacy ‘’Grigore T. Popa‘’, Iasi. The inclusion criteria for patients with prediabetes were an HbA1c of 5.7–6.4% and an FPG of 110–125 mg/dL; for patients with T2DM, the inclusion criteria were an HbA1c ≥6.5% and an FPG ≥126 mg/dL. The exclusion criteria were chronic gastrointestinal disease, systemic antibiotics within 6 weeks before inclusion, the use of probiotics within 3 months before inclusion, regular intake of insulin or insulin analogs, antibiotics or probiotics, antacids, H2-receptor blockers, proton pump inhibitors, loperamide, cholestryramine, ω3-unsaturated fatty acid supplements, fibrates, corticosteroids or sex steroids, significant immunodeficiency, breast-feeding or pregnancy, psychiatric illness under the care of a psychiatrist, eating disorders such as bulimia, patients on special diets, hypothalamic or genetic etiology of obesity, a current diagnosis of cancer, any surgery planned in the ensuing 6 months, substance abuse, use of prescription medications or over-the-counter drugs affecting metabolism, excessive intake of alcohol, excessive intake of caffeine, and an inability or unwillingness to comply with the protocol.

### 2.2. Assessment of Dietary Intake

The food information was collected using the EPIC-Norfolk FFQ (food frequency questionnaire), which was previously validated for the Romanian population [12]. The questionnaire comprised 14 food groups that contained a total of 130 items, both food and beverages which are shown in Appendix A. For each of the 130 items, the participants indicated the frequency of consumption by choosing one of the following options: 1 (never or less than once/month), 2 (1–3 per month), 3 (once a week), 4 (2–4 per week), 5 (5–6 per week), 6 (once a day), 7 (2–3 per day), 8 (4–5 per day) or 9 (6+ per day). A medium-sized portion was assigned to each item in the questionnaire, and the portion was expressed either as common portions, such as an apple or a slice of bread, or using household measurements, such as a glass or spoon.

The data collected using the food frequency questionnaire were analyzed using the FETA-FFQ EPIC Tool, which generated an output containing the average daily intake of nutrients and 14 basic food groups, such as alcoholic beverages; cereals and cereal products; eggs and egg dishes; fats and oils; fish and fish products; fruit; meat and meat products; milk and milk products; non-alcoholic beverages; nuts and seeds; potatoes; soups and sauces; sugars; preserves and snacks; and vegetables.

### 2.3. Identification of Dietary Patterns

We conducted a factor analysis (principal component) on the 14 food groups, followed by varimax orthogonal rotation, to identify dietary patterns. Bartlett’s sphericity test and the Kaiser–Meyer–Olkin coefficient (KMO) were carried out at a cut-off value of *p* ≤ 0.05 and a KMO ≥ 0.50, respectively, indicating a satisfactory level of confidence for the factorial analysis. The criteria that determined the number of factors retained were the components with eigenvalues higher than 1.3, the Cattel’s scree plot, in which the number of points on the more declined determine the appropriate number of factors to be retained, and the conceptual significance of the patterns was identified. After establishing the factors, they were rotated with the varimax method. The principal components were determined as the foods that presented important factor loading, which was established at a level ≥ 0.2 (indicating direct correlation with the pattern) or ≤ −0.2 (inverse correlation with the pattern). Each individual received a score for each principal component. The factor scores were derived by multiplying each factor loading by the corresponding food group value for the individual, and then summing across food groups to determine the participant’s factor score for each pattern.

### 2.4. Blood Pressure and Anthropometric Measurements

For the blood pressure (BP) and heart rate measurements, the participants were first asked to rest in the sitting position for five to ten minutes. The BP and heart rate were measured from the non-dominant arm using a standardized, calibrated Omron M2 sphygmomanometer. The height was measured for all patients (barefoot with head upright) and was reported to the nearest 0.1 cm. The weight in light clothes was measured using a standard scale (Omron HN 288) that was placed on a hard-floor and was reported to the nearest 0.1 kg. The body mass index (BMI) was calculated by dividing the weight in kilograms by the square of the height in meters. The body fat (BF) percentage was calculated using the Omron HBF-306C body analyzer. The waist circumference (WC) was measured at the approximate midpoint between the lower margin of the last palpable rib and the top of the iliac crest, using a stretch-resistant tape. All measurements were taken by a trained dietitian using standardized procedures.

### 2.5. Biochemical Analyses

The blood samples were obtained after 12 h of overnight fasting and were collected from all patients by trained nurses. Venous blood was drawn into vacutainer tubes and used for blood chemistry analyses. Serum was separated immediately, and the extracted serum was analyzed for fasting plasma glucose (FPG), HbA1c, serum insulin, total cholesterol, triglycerides, HDL and LDL cholesterol. The laboratory tests were conducted in a private licensed laboratory.

### 2.6. Data Analyses

Based on the dietary pattern scores, the study participants were classified according to tertiles. The three tertiles indicated adherence to the pattern. Patients in tertile 1 (T1) had low adherence, those in tertile 2 (T2) had medium adherence, and those in tertile 3 (T3) had high adherence to the dietary pattern. This allows for the evaluation of the effects of dietary adherence on blood pressure, anthropometric parameters, and the glycemic and lipid profiles [13]. Significant differences between the prediabetic and diabetic groups were assessed with the *t*-test for continuous variables and the chi-square test for categorical variables. Significant differences between tertile categories were assessed with one-way ANOVA with Tukey’s post hoc comparisons. The relationships between food groups intake and anthropometric measurements and biomarkers were assessed using Pearson or Spearman correlation coefficients. A *p*-value < 0.05 was considered statistically significant. The data are presented as means ± standard errors of the mean (SEM) for continuous variables or sum (percentage) of patients for categorical variables. To address the impact of the tertiles on metabolic control of diabetes (HbA1c > 7%), the relative risk (RR) was estimated as odds ratios (OR), with tertile 1 used as a reference. All statistical analyses were performed with SPSS 20 (SPSS Inc., Chicago, IL, USA).

## 3. Results

The general characteristics, anthropometric measurements and biomarkers of the study participants from the two groups are presented in Table 1. Compared with prediabetic patients, patients with T2DM had significantly higher systolic blood pressure (SBP) (145.4 ± 0.8 vs. 142.1 ± 1.3 mm Hg; *p* = 0.033), WC (106.1 ± 0.5 vs. 102.7 ± 0.9 cm; *p* = 0.001), FPG (159.5 ± 2.4 mg/dL vs. 122.1 ± 2.1 mg/dL; *p* = 0.000), HbA1c (7.8 ± 0.1 vs. 5.8 ± 0.1 %; *p* = 0.000), total cholesterol (201.5 ± 2.1 vs. 189.8 ± 2.9 mg/dL; *p* = 0.002), triglycerides (166.5 ± 5.1 vs. 135.9 ± 5.8 mg/dL; *p* = 0.000), LDL cholesterol (120.6 ± 1.6 vs. 113.1 ± 2.2 mg/dL; *p* = 0.000), and a significantly lower HDL cholesterol (51.8 ± 0.7 vs. 55.1 ± 0.9 mg/dL; *p* = 0.010).

In terms of food intake, patients with T2DM had a significantly higher intake of fish and fish products (22.8 ± 1.1 vs. 19.4 ± 1.2 g; *p* = 0.047), fruit (287.7 ± 9.3 vs. 243.3 ± 9.9 g; *p* = 0.001) and a significantly lower intake of fats and oils (6.4 ± 0.2 vs. 7.8 ± 0.5 g; *p* = 0.018), potatoes (57.1 ± 1.7 vs. 66.6 ± 4.8 g; *p* = 0.024), sugars, preserves and snacks (17.0 ± 0.8 vs. 20.6 ± 1.3 g; *p* = 0.021) compared to prediabetic patients (Table 2).

After confirming the adequacy of the data via the KMO coefficient and Bartlett’s sphericity test, the 14 food groups that were included in the analysis resulted in three identified dietary patterns, which together explained 39.5% of the total variance in intake in the prediabetic group. Pattern 1 explained 20.0% of the dietary intake variance, and was composed of fats and oils, fruit, cereals and cereal products, sugars, preserves and snacks, non-alcoholic beverages, nuts and seeds, with a negative load for soups and sauces. Pattern 2 explained 9.9% of the dietary intake variance, and included fats and oils, fruit, cereals and cereal products, potatoes, soups and sauces, vegetables, milk and milk products, and fish and fish products. Pattern 3 explained 9.6% of the dietary intake variance, and was composed of sugars, preserves and snacks, soups and sauces, vegetables, meat and meat products, alcoholic beverages, and eggs and eggs dishes.

Within the group of patients with T2DM, the three identified dietary patterns explained 37.4% of the total variance in intake. Pattern 1 explained 16.2% of the dietary intake variance, and included fats and oils, cereals and cereal products, potatoes, sugars, preserves and snacks, and eggs and egg dishes, with a negative load for soups and sauces. Pattern 2 explained 11.0% of the dietary intake variance, and was composed of sugars, preserves and snacks, fruit, vegetables, non-alcoholic beverages, nuts and seeds, fish and fish products, and milk and milk products. Pattern 3 explained 10.2% of the dietary intake variance, and was composed of potatoes, vegetables, meat and meat products, soups and sauces, eggs and egg dishes, and milk and milk products (Table 3).

General characteristics, anthropometric data and biomarkers of prediabetic patients across the tertiles of the main dietary pattern scores are shown in Table 4. Overall, of the patients from Pattern 1, T1 had the lowest SBP, DBP, body weight, WC, FPG, HbA1c and triglycerides, and the highest average HDL cholesterol. Moreover, T1 patients had a significantly lower FPG than those in T3 (114.6 ± 3.9 vs. 125.4 ± 3.4 mg/dL; *p* = 0.044). In Pattern 2, the patients in T1 were younger than those in T2 (56.2 ± 1.8 vs. 61.8 ± 1.4 years; *p* = 0.029) and T3 (56.2 ± 1.8 vs. 61.0 ± 0.9 years; *p* = 0.036), and had a lower SBP (133.4 ± 2.9 vs. 146.4 ± 2.9 mm Hg; *p* = 0.002/133.4 ± 2.9 vs. 143.9 ± 1.6 mm Hg; *p* = 0.004), FPG (105.5 ± 3.4 vs. 125.1 ± 4.4 mg/dL; *p* = 0.001/105.5 ± 3.4 vs. 124.8 ± 3.1 mg/dL; *p* = 0.000), serum insulin (9.5 ± 0.7 vs. 14.4 ± 1.3 uIU/mL; *p* = 0.005/9.5 ± 0.7 vs. 14.9 ± 1.2 uIU/mL; *p* = 0.001) compared to those in T2 and T3, respectively. On the other hand, patients in T2 had the highest weight, WC, serum insulin and triglycerides.

The general characteristics, antropometric data and biomarkers of T2DM patients across tertiles of the main dietary patterns are shown in Table 5. In Pattern 1, patients in T1 had a significantly lower SBP than those in T3 (141.4 ± 1.3 vs. 147.2 ± 1.3 mm Hg; *p* = 0.021). Furthermore, in T1, Pattern 1 had the lowest SBP, DBP, body weight and BMI, although these were not statistically different compared to those of the other tertiles. In Pattern 3, T1 had a significantly lower Hb1Ac than those in T2 and T3 (7.5 ± 0.1 vs. 7.9 ± 0.1%; *p* = 0.036), but a higher LDL cholesterol than patients in T2 (125.1 ± 3.3 vs. 113.8 ± 3.3 mg/dL; *p* = 0.041).

Food intake. The analysis of food consumption by food groups showed that, compared with prediabetic patients, diabetic patients consumed the following foods with lower frequencies: beef burgers (1.1 ± 0.3 vs. 1.2 ± 0.4; *p* = 0.002); sausages (1.7 ± 1.0 vs. 2.0 ± 1.1; *p* = 0.000); savory pies (1.1 ± 0.3 vs. 1.3 ± 0.7; *p* = 0.002); breakfast cereal such as corn flakes and muesli (1.4 ± 0.9 vs. 1.6 ± 1.1; *p* = 0.020); chips (2.0 ± 1.0 vs. 2.2 ± 1.0; *p* = 0.009); lasagna and moussaka (1.1 ± 0.4 vs. 1.3 ± 0.5; *p* = 0.000); pizza (1.4 ± 0.6 vs. 1.6 ± 0.7; *p* = 0.000); salad, cream and mayonnaise (1.1 ± 0.4 vs. 1.2 ± 0.5; *p* = 0.026); home baked fruit pies (1.6 ± 0.7 vs. 1.7 ± 0.7; *p* = 0.036); home baked cakes (1.2 ± 0.4 vs. 1.3 ± 0.6; *p* = 0.018); milk puddings (1.4 ± 0.8 vs. 1.6 ± 0.8; *p* = 0.016); chocolate snack bars (1.1 ± 0.5 vs. 1.3 ± 0.7; *p* = 0.015); sugar added to tea, coffee and cereal (2.1 ± 1.9 vs. 2.9 ± 2.2; *p* = 0.000); crisps or other packet snacks (1.3 ± 0.7 vs. 1.5 ± 0.9; *p* = 0.001); sauces (1.2 ± 0.6 vs. 1.3 ± 0.7; *p* = 0.011); tomato ketchup (2.2 ± 1.4 vs. 2.7 ± 1.7; *p* = 0.000); peaches, plums and apricots (3.1 ± 1.3 vs. 3.3 ± 1.3; *p* = 0.020); tinned fruit (1.5 ± 0.9 vs. 1.7 ± 0.9; *p* = 0.031); leeks (1.3 ± 0.7 vs. 1.5 ± 1.0; *p* = 0.002); sweetcorn (1.4 ± 0.7 vs. 1.5 ± 1.0; *p* = 0.040); baked beans (1.7 ± 0.6 vs. 1.8 ± 0.6; *p* = 0.011); and tofu (1.2 ± 0.4 vs. 1.3 ± 0.5; *p* = 0.002). On the other hand, the patients with T2DM consumed the following more frequently than patients with prediabetes: fried fish in butter, as in fish and chips (1.5 ± 0.8 vs. 1.4 ± 0.7; *p* = 0.031); wine (1.7 ± 1.2 vs. 1.5 ± 1.0; *p* = 0.017); apples (4.9 ± 1.4 vs. 4.6 ± 1.5; *p* = 0.002); grapes (2.6 ± 1.4 vs. 2.3 ± 1.1; *p* = 0.002); parsnips, turnips and swedes (3.1 ± 1.7 vs. 2.4 ± 1.3; *p* = 0.000); onions (5.1 ± 1.0 vs. 4.8 ± 1.2; *p* = 0.003); sweet peppers (3.1 ± 1.2 vs. 2.9 ± 1.1; *p* = 0.031); and green salad, lettuce, cucumber and celery (3.2 ± 1.5 vs. 2.9 ± 1.3; *p* = 0.005) (Appendix A).

In the prediabetic group, direct correlations were observed between body weight and the intake of meat and meat products (r = 0.142, *p* = 0.001); WC and the intake of meat and meat products (r = 0.127, *p* = 0.002); FPG and potatoes intake (r = 0.151, *p* = 0.000); FPG and soups and sauces intake (r = 0.086, *p* = 0.048); HbA1c and meat and meat products intake (r = 0.119, *p* = 0.042); serum, insulin and milk and milk products intake (r = 237, *p* = 0.000); and between LDL cholesterol and the intake of meat and meat products (r = 0.086, *p* = 0.037). Inverse correlations were observed between BF and intake of cereals and cereal products (r = −0.127, *p* = 0.003); HbA1c and the intake of nuts and seeds (r = −0.116, *p* = 0.039); and HDL cholesterol and eggs and egg dishes intake (r = −0.088, *p* = 0.032) (Appendix A).

In T2DM patients, there were direct correlations found between body weight and fruit intake (r = 0.134, *p* = 0.030); FPG and fruit intake (r = 0.184, *p* = 0.003); FPG and milk and milk products (r = 0.190, *p* = 0.002); triglycerides and milk and milk products (r = −0.138, *p* = 0.025); and there were inverse correlations between BF and the intake of cereals and cereal products (r =−0.147, *p* = 0.018) and between HbA1c and fats and oils intake (r = −0.167, *p* = 0.007) (Appendix A). In T2DM patients, poor metabolic control of diabetes was 1.8 (95% CI: 0.56, 5.74) times more likely to occur in tertile 3 than in tertile 1 of Pattern 1. Moreover, a high adherence to Pattern 2 was associated with an increased RR of 1.1 (95% CI: 0.37, 3.46). On the other hand, the risk for poor metabolic control of diabetes was reduced by 0.75 times (95% CI: 0.27, 2.10) in Pattern 3.

## 4. Discussion

In the present study, we evaluated dietary patterns of prediabetic and T2DM patients and compared their dietary intake. Using factorial analyses, three dietary patterns were identified and described for the two groups of patients. When comparing the degree of adherence to a specific dietary pattern by tertiles, several important results were observed. In the prediabetic group, patients with a low adherence to Pattern 2 had a lower SBP, FPG and serum insulin than those with high adherence. This dietary pattern was characterized by a high intake of potatoes, soups and sauces, vegetables, cereals and cereal products, fruit, fats and oils, milk and milk products, and fish and fish products. These results are consistent with those reported by Montonen et al. [14], who showed that a dietary pattern consisting of butter, potatoes and whole milk was positively correlated with the risk of diabetes. Several studies reported an association between the increased risk of hypertension and T2DM, and a high consumption of baked, boiled, mashed or fried potatoes [15,16]. Not surprisingly, dietary recommendations for patients with diabetes include reduced starchy foods with a high glycemic index, such as potatoes, white rice and white bread.

Furthermore, patients with a low adherence to Pattern 1 had a lower FPG than those with a high adherence. Pattern 1 was characterized by a high intake of fats and oils, fruit, cereals and cereal products, sugars, preserves and snacks, non-alcoholic beverages, nuts and seeds, and a low intake of soups and sauces. These findings are in line with the results of epidemiological studies indicating that a high intake of saturated fats was associated with a higher level of fasting glucose. Dietary recommendations for patients with diabetes include increased consumption of whole grains, and decreased consumption of refined cereals, given that refined cereal products have a high glycemic index, while whole grains have a moderate glycemic index [17] and a high content of dietary fiber, antioxidants and phytochemicals [18]. Thus, consumption of refined grains, white bread, and sugar and sugar-containing soft drinks could explain the increased blood glucose in tertile 3 compared to tertile 1. The intake of nuts and seeds did not seem to play a significant role in increasing blood glucose in this dietary pattern, although they have been associated with improvements in plasma insulin, insulin resistance and blood glucose [19].

In the T2DM group, we found that patients in tertile 1 of Pattern 1 had a significantly lower SBP than patients in tertile 3. Sabour et. al. demonstrated that high cholesterol and saturated fat intake is associated with hypertension [20]. High intakes of salt and sugar (mainly fructose from added sugars) have been linked to the etiology of hypertension, and this may be especially true for countries in food transition [21]. The results of a meta-analysis indicated that there were no significant effects of egg consumption on BP [22]; however, other studies showed that egg and cholesterol intake was associated with an increased risk of hypertension in French women [23], which varied on whether the consumed eggs were boiled, scrambled or fried. Furthermore, the patients in tertile 1 of dietary Pattern 3 had significantly lower HbA1c levels. This dietary pattern was characterized by a high intake of meat and meat products, soups and sauces, eggs and egg dishes, vegetables, milk and milk products and potatoes. Thus, a low adherence to this dietary pattern is beneficial, since an increased risk for prediabetes and diabetes has been associated with a Western diet [24].

In terms of dietary intake, there was a lower consumption of fats and oils, potatoes, sugar, preserves and snacks, and a higher consumption of fish, fish products and fruit, in patients with T2DM compared to patients with prediabetes. Nutrition guidelines for patients with diabetes recommend elimination of trans fats and limiting the consumption of saturated fats; reducing the consumption of potatoes; replacing potatoes with brown rice and other cereals; and avoiding sugar. Other recommendations for patients with T2DM include eating nutrient-dense sources of carbohydrates that are high in fiber, such as vegetables and fruits; replacing red meat with beans, nuts, fish and skinless chicken; and consuming one or more sources of omega 3 daily, such as fatty fish, nuts and soybean oil [17]. Furthermore, we showed that, compared with prediabetic patients, T2DM patients consumed processed meat less frequently (beef burgers, sausages), fast food products (savory pies, chips, pizza) or foods that are high in sugar (fruit pies, cakes, milk puddings, chocolate snack bars, sugar, tinned fruit). Foods that were frequently consumed by T2DM patients were apples and vegetables (parsnips, turnips, swedes, onions, sweet peppers, green salad, cucumber). These differences in consumption between prediabetic and T2DM patients may be, in part, the result of nutritional medical therapy that the T2DM patients received at each regular medical visit.

Correlational analyses in patients with prediabetes showed a significant difference between the intake of meat and meat products, body weight and WC. These findings are consistent with those of Wang et. al. [25], who reported positive associations between meat consumption, obesity and abdominal obesity. Moreover, the results of a systematic review indicated that there was a direct association between the consumption of red meat and processed meat, with an increased risk of obesity, a high BMI and an increased WC [26]. Meat consumption may be associated with increased risk for obesity because it is high in calories and fat content [25]. Furthermore, within this group, an inverse correlation was observed between nut consumption and HbA1c. A systematic review that included patients with T2DM showed that consumption of nuts improved HbA1c and blood glucose [27]. Nuts are rich sources of monounsaturated and polyunsaturated fatty acids and vegetable protein, and their inclusion in a diet significantly improves its nutritional quality [28]. Moreover, an inverse correlation was observed between the intake of eggs and egg preparations, and the level of HDL-cholesterol; however, more studies reported that egg consumption has a beneficial effect on HDL-cholesterol [29]. The difference in results could be explained by the consumption of fried eggs by patients in the study. Similar results in rats demonstrated that consumption of fried oil decreased HDL-cholesterol levels [30].

Within the group with T2DM, a direct correlation was observed between body weight and fruit consumption. This result is different from those of other studies [31,32], and may be due to the type of fruit consumed. For example, canned fruits often contain added sugar, and may also contain lower concentrations of heat-sensitive nutrients following the canning procedure [33]. One study showed that participants who reported more frequent consumption of canned fruit had an increased risk of mortality. Moreover, replacing non-preserved fruit with preserved fruit was associated with a modest increase in mortality risk [33]. In addition, a direct correlation was observed between the level of triglycerides and the intake of milk and dairy products in T2DM patients. These findings are somewhat different from those of other studies, which demonstrated a positive association between the intake of dairy products and a decrease in the level of triglycerides [34,35]. However, whole milk products are an important source of saturated fatty acids [36], and a high intake of saturated fatty acids and a low intake of monounsaturated and polyunsaturated fatty acids are closely correlated with increased levels of triglycerides in the blood [36]. We also showed that there was an inverse correlation between the intake of fats and oils and HbA1c levels. The results of a meta-analysis showed that olive oil consumption could be beneficial for the prevention and management of T2DM [37]. On the other hand, the consumption of lard, peanut oil, and other types of refined oils has been associated with an increased risk for T2DM [38]. It should not be overlooked that fat ingestion before carbohydrate meals slows gastric emptying, thereby attenuating the rapid postprandial rise in blood glucose [39], which could, in part, explain this correlation.

We also observed an inverse correlation between cereals and cereal products intake and BF percentage in both groups. These results are consistent with the results of McKeown et al., who demonstrated that a high intake of fiber from cereals, mainly from whole grains, was associated with a decrease in BF percentage [40]. Bazzano et al. [41] reported an inverse association between the breakfast intake of whole and refined grains and weight in a cohort of 17,881 men. Likewise, other studies showed an inverse association between the intake of whole grains rich in fiber and weight gain, and a positive association between the consumption of refined grains and weight gain [42], whereas whole grain intake was inversely associated with weight gain [43]. It is important to note that white bread was the most consumed product from this food group, but the patients in this study also consumed other products, such as whole grain bread, oatmeal, corn flakes and muesli; therefore, this correlation can be mainly attributed to the consumption of whole grains.

Food patterns are crucial in the development of dietary guidelines, and the results obtained in this study can be used in the formulation and development of practical guidelines and policies to improve nutrition and prevent the occurrence of prediabetes and type 2 diabetes, and in the management of these conditions. FFQs are convenient and cost-effective, but they are also subjective and prone to measurement errors, since they rely on respondents’ memory and estimation of usual food portion sizes. Thus, the data accuracy may suffer, including overestimating or underestimating food intake, which represents a limitation of the present study.

## 5. Conclusions

The study demonstrated that certain food patterns were associated with higher blood pressure, fasting blood glucose and serum insulin. Prediabetic patients with a high adherence to Pattern 1, which was mainly composed of fats and oils, fruits, cereals and cereal products, sugar, preserves and snacks, had a higher FPG. Furthermore, increased adherence to Pattern 2, composed mainly of potatoes, soups and sauces, vegetables and cereals and cereal products, increased SBP, FPG and serum insulin levels. T2DM patients with increased adherence to Pattern 3, which was mainly composed of meat and meat products, soups and sauces, eggs and egg dishes, showed an increased risk for high HbA1c levels. Therefore, our findings demonstrate that more research is needed to better clarify links between dietary patterns and the prevention and control of prediabetes and diabetes.

## Figures and Tables

**Table 1 metabolites-13-00532-t001:** General characteristics, anthropometric measurements and biomarkers of study participants.

Variables	Prediabetic (*n* = 264)	T2DM (*n* = 587)	*p* Value
Age (years)	60.1 ± 0.7 ^a^	62.3 ± 0.4 ^b^	0.007
Men n, (%)	92 (34.8%)	232 (39.5%)	0.194
Women n, (%)	172 (65.2%)	355 (60.5%)	
Urban n, (%)	165 (62.5%)	349 (59.5%)	0.506
Rural n, (%)	99 (37.5%)	238 (40.5%)	
SBP (mm Hg)	142.1 ± 1.3 ^a^	145.4 ± 0.8 ^b^	0.033
DBP (mm Hg)	86.0 ± 0.7	86.5 ± 0.4	0.524
Weight (kg)	86.4 ± 1.0	87.7 ± 0.6	0.304
WC (cm)	102.7 ± 0.9 ^a^	106.1 ± 0.5 ^b^	0.001
BMI (kg/m^2^)	31.4 ± 0.3	31.6 ± 0.2	0.612
BF (%)	35.3 ± 0.5	35.6 ± 0.4	0.809
FPG (mg/dL)	122.1 ± 2.1 ^a^	159.5 ± 2.4 ^b^	0.000
HbA1c (%)	5.8 ± 0.1 ^a^	7.8 ± 0.1 ^b^	0.000
Serum insulin (µIU/mL)	13.5 ± 0.7	14.1 ± 0.5	0.524
Total cholesterol (mg/dL)	189.8 ± 2.9 ^a^	201.5 ± 2.1 ^b^	0.002
Triglycerides (mg/dL)	135.9 ± 5.8 ^a^	166.5 ± 5.1 ^b^	0.000
HDL cholesterol (mg/dL)	55.1 ± 0.9 ^a^	51.8 ± 0.7 ^b^	0.010
LDL cholesterol (mg/dL)	113.1 ± 2.2 ^a^	120.6 ± 1.6 ^b^	0.000

Categorical variables are presented as sums and percentages, and continuous variables are presented as means ± SEMs; ^a,b^ significantly different between groups; SBP, systolic blood pressure; DBP, diastolic blood pressure; WC, waist circumference; BMI, body mass index; BF, body fat; FPG, fasting plasma glucose; HbA1c, glycated hemoglobin; HDL, high-density lipoprotein; LDL, low-density lipoprotein.

**Table 2 metabolites-13-00532-t002:** Food group intake of study participants.

Variables	Prediabetic	T2DM	*p* Value
Alcoholic beverages (g)	20.9 ± 4.0	21.0 ± 2.2	0.984
Cereals and cereal products (g)	200.6 ± 7.5	192.2 ± 3.7	0.265
Eggs and egg dishes (g)	20.8 ± 1.3	21.1 ± 0.6	0.809
Fats and oils (g)	7.8 ± 0.5 ^a^	6.4 ± 0.2 ^b^	0.018
Fish and fish products (g)	19.4 ± 1.2 ^a^	22.8 ± 1.1 ^b^	0.047
Fruit (g)	243.3 ± 9.9 ^a^	287.7 ± 9.3 ^b^	0.001
Meat and meat products (g)	109.1 ± 4.2	103.5 ± 2.5	0.241
Milk and milk products (g)	305.5 ± 14.2	302.3 ± 9.3	0.849
Non-alcoholic beverages (g)	334.9 ± 13.2	337.0 ± 9.7	0.903
Nuts and seeds (g)	4.3 ± 0.7	3.9 ± 0.3	0.514
Potatoes (g)	66.6 ± 4.8 ^a^	57.1 ± 1.7 ^b^	0.024
Soups and sauces (g)	223.7 ± 8.4	223.0 ± 5.5	0.943
Sugars, preserves and snacks (g)	20.6 ± 1.3 ^a^	17.0 ± 0.8 ^b^	0.021
Vegetables (g)	260.8 ± 9.0	276.8 ± 5.9	0.137

Variables are presented as means ± SEMs; ^a,b^ significantly different between groups.

**Table 3 metabolites-13-00532-t003:** Factor loading matrix for main dietary patterns of prediabetic and type 2 diabetes patients.

	Prediabetic			T2DM		
Food Group	Pattern 1	Pattern 2	Pattern 3	Pattern 1	Pattern 2	Pattern 3
Fats and oils	0.700	0.371	-	0.821	-	-
Fruit	0.653	0.375	-	-	0.671	-
Cereals and cereal products	0.652	0.424	-	0.651	-	-
Sugars, preserves and snacks	0.576	-	0.552	0.540	0.321	-
Non-alcoholic beverages	0.448	-	-	-	0.535	-
Nuts and seeds	0.256	-	-	-	0.456	-
Potatoes	-	0.656	-	0.616	-	0.205
Soups and sauces	−0.293	0.616	0.203	−0.226	-	0.670
Vegetables	-	0.616	0.238	-	0.667	0.335
Milk and milk products	-	0.358	-	-	0.236	0.272
Fish and fish products	-	0.297	-	-	0.216	-
Meat and meat products	-	-	0.769	-	-	0.720
Alcoholic beverages	-	-	0.686	-	-	-
Eggs and egg dishes	-	-	0.238	0.293	-	0.463
Variance explained (%)	20.0	9.9	9.6	16.2	11.0	10.2

Food groups with factor loadings ≥ 0.2 and ≤ −0.2.

**Table 4 metabolites-13-00532-t004:** General characteristics, anthropometric measurements and biomarkers of prediabetic patients across tertiles of dietary patterns.

Variables	T1	Pattern 1 T2	T3	T1	Pattern 2 T2	T3	T1	Pattern 3 T2	T3
Age (years)	59.6 ± 1.5	60.1 ± 1.5	60.3 ± 1.1	56.2 ± 1.8 ^c^	61.8 ± 1.4 ^d^	61.0 ± 0.9 ^d^	59.7 ± 1.6	60.6 ± 1.5	60.1 ± 1.0
SBP (mm Hg)	139.0 ± 3.4	142.2 ± 2.2	143.5 ± 1.7	133.4 ± 2.9 ^e^	146.4 ± 2.9 ^f^	143.9 ± 1.6 ^f^	143.7 ± 2.8	140.6 ± 2.2	142.1 ± 1.9
DBP (mm Hg)	84.0 ± 1.4	86.4 ± 1.5	86.8 ± 0.9	82.4 ± 1.7 ^g^	88.5 ± 1.4 ^h^	86.4 ± 0.8 ^gh^	87.8 ± 1.7	85.9 ± 1.3	85.3 ± 0.9
Weight (kg)	84.4 ± 2.2	87.5 ± 2.1	86.9 ± 1.4	84.3 ± 2.4	86.9 ± 1.7	87.2 ± 1.5	84.7 ± 2.0	88.4 ± 2.2	86.3 ± 1.4
WC (cm)	102.8 ± 1.7	103.3 ± 1.6	102.4 ± 1.3	99.3 ± 2.0	103.0 ± 1.4	104.2 ± 1.2	99.8 ± 1.8	104.2 ± 1.7	103.4 ± 1.2
BMI (kg/m^2^)	31.3 ± 0.8	31.6 ± 0.7	31.4 ± 0.4	30.9 ± 0.7	31.2 ± 0.5	31.8 ± 0.5	31.0 ± 0.7	31.6 ± 0.7	31.6 ± 0.5
BF (%)	35.5 ± 0.9	36.0 ± 1.0	34.8 ± 0.7	35.4 ± 1.0	35.5 ± 0.9	35.1 ± 0.7	35.9 ± 0.9	35.1 ± 1.0	35.1 ± 0.7
FPG (mg/dL)	114.6 ± 3.9 ^a^	119.1 ± 3.5 ^ab^	125.4 ± 3.4 ^b^	105.5 ± 3.4 ^i^	125.1 ± 4.4 ^j^	124.8 ± 3.1 ^j^	118.2 ± 3.5	125.0 ± 4.6	122.6 ± 3.2
HbA1c (%)	5.8 ± 0.1	5.8 ± 0.1	5.8 ± 0.1	5.7 ± 0.1	5.8 ± 0.1	5.8 ± 0.0	5.8 ± 0.1	5.7 ± 0.1	5.8 ± 0.1
Serum insulin (uIU/mL)	11.1 ± 0.9	14.3 ± 1.5	14.3 ± 1.2	9.5 ± 0.7 ^k^	14.4 ± 1.3 ^m^	14.9 ± 1.2 ^m^	11.2 ± 1.0	15.0 ± 1.2	13.8 ± 1.2
Total cholesterol (mg/dL)	193.8 ± 5.3	190.6 ± 5.5	187.3 ± 4.3	194.7 ± 6.4	186.7 ± 5.1	189.1 ± 4.1	188.2 ± 6.0	186.7 ± 5.0	192.1 ± 4.3
Triglycerides (mg/dL)	128.8 ± 9.9	124.6 ± 9.0	145.2 ± 9.6	120.0 ± 8.6	139.0 ± 10.9	141.8 ± 9.3	123.9 ± 7.9	152.8 ± 12.2	133.5 ± 9.1
HDL cholesterol (mg/dL)	55.9 ± 2.1	56.2 ± 1.9	54.2 ± 1.3	57.9 ± 1.9	57.1 ± 2.1	52.9 ± 1.3	55.4 ± 1.7	55.3 ± 2.1	54.9 ± 1.4
LDL cholesterol (mg/dL)	114.1 ± 4.4	116.0 ± 4.1	111.2 ± 3.3	117.5 ± 4.7	105.4 ± 4.0	115.2 ± 3.2	113.4 ± 4.4	106.0 ± 3.7	116.5 ± 3.4

Tertiles of dietary pattern scores. Continuous variables are presented as means ± SEMs; SBP, systolic blood pressure; DBP, diastolic blood pressure; WC, waist circumference; BMI, body mass index; BF, body fat; FPG, fasting plasma glucose; HbA1c, glycated hemoglobin; HDL, high-density lipoprotein; LDL, low-density lipoprotein.; ^a,b^ significantly different between T1 and T3, Pattern 1; ^c,d^ significantly different between T1 and T2 of Pattern 2; ^e,f^ significantly different between T1, T2 and T3 of Pattern 2; ^g,h^ significantly different between T1 and T2 of Pattern 2; ^i,j^ significantly different between T1, T2 and T3 of Pattern 2; ^k,m^ significantly different between T1, T2 and T3 of Pattern 2.

**Table 5 metabolites-13-00532-t005:** General characteristics, anthropometric measurements and biomarkers of T2DM patients across tertiles of dietary patterns.

	T1	Pattern 1 T2	T3	T1	Pattern 2 T2	T3	T1	Pattern 3 T2	T3
Age (year)	61.9 ± 0.8	62.9 ± 0.7	62.1 ± 0.5	61.5 ± 0.8	63.7 ± 0.6	62.0 ± 0.5	62.0 ± 0.9	62.2 ± 0.8	62.4 ± 0.5
SBP (mm Hg)	141.4 ± 1.3 ^a^	146.0 ± 1.6 ^ab^	147.2 ± 1.3 ^b^	143.7 ± 1.3	145.2 ± 1.5	146.4 ± 1.3	142.8 ± 1.4	144.9 ± 1.5	147.6 ± 1.2
DBP (mm Hg)	85.5 ± 0.9	86.0 ± 1.1	87.4 ± 0.8	86.4 ± 1.0	86.7 ± 1.0	86.6 ± 0.8	85.6 ± 0.8	85.6 ± 0.9	87.8 ± 0.8
Weight (kg)	85.8 ± 1.3	88.7 ± 1.4	88.2 ± 0.9	87.3 ± 1.2	86.9 ± 1.3	88.3 ± 0.9	86.0 ± 1.2	87.2 ± 1.3	88.8 ± 0.9
WC (cm)	105.0 ± 1.0	106.6 ± 1.2	106.5 ± 0.7	104.7 ± 1.0	106.2 ± 0.9	106.8 ± 0.8	104.4 ± 1.0	105.4 ± 1.1	107.4 ± 0.7
BMI (kg/m^2^)	31.3 ± 0.4	32.4 ± 0.4	31.5 ± 0.3	31.5 ± 0.5	31.4 ± 0.4	31.9 ± 0.3	31.9 ± 0.4	31.4 ± 0.4	31.7 ± 0.3
BF (%)	35.8 ± 0.6	36.9 ± 1.3	34.7 ± 0.4	35.8 ± 0.6	35.3 ± 0.6	35.6 ± 0.7	36.2 ± 0.7	35.4 ± 0.6	35.3 ± 0.7
FPG (mg/dL)	153.0 ± 4.7	160.4 ± 4.8	162.3 ± 3.4	154.9 ± 4.8	159.8 ± 4.5	161.7 ± 3.5	151.1 ± 4.3	161.7 ± 5.1	162.7 ± 3.4
HbA1c (%)	7.8 ± 0.1	7.7 ± 0.1	7.9 ± 1.2	7.8 ± 0.1	7.6 ± 0.1	7.9 ± 0.1	7.5 ± 0.1 ^c^	7.9 ± 0.1 ^d^	7.9 ± 0.1 ^d^
Serum insulin (uIU/mL)	14.5 ± 0.9	15.5 ± 1.3	13.1 ± 0.5	13.7 ± 0.9	15.4 ± 1.2	13.5 ± 0.6	12.9 ± 0.9	14.8 ± 1.2	14.2 ± 0.6
Total cholesterol (mg/dL)	202.4 ± 4.3	201.8 ± 4.3	200.8 ± 3.1	201.2 ± 3.9	197.9 ± 4.2	203.4 ± 3.2	206.3 ± 4.3	192.7 ± 4.1	203.4 ± 3.1
Triglycerides (mg/dL)	174.2 ± 15.8	157.7 ± 6.4	167.1 ± 5.7	153.7 ± 6.3	167.7 ± 8.9	172.3 ± 8.6	158.6 ± 6.4	163.4 ± 7.7	172.0 ± 8.9
HDL cholesterol (mg/dL)	51.2 ± 1.3	51.6 ± 1.2	52.2 ± 1.0	50.9 ± 1.1	52.0 ± 1.4	52.1 ± 1.0	52.3 ± 1.3	51.7 ± 1.3	51.6 ± 1.0
LDL cholesterol (mg/dL)	119.9 ± 3.1	121.5 ± 3.1	120.4 ± 2.4	122.4 ± 3.0	116.5 ± 3.3	121.7 ± 2.3	125.1 ± 3.3 ^e^	113.8 ± 3.3 ^f^	121.7 ± 2.2 ^ef^

Tertiles of dietary pattern scores. Continuous variables are presented as means ± SEMs; SBP, systolic blood pressure; DBP, diastolic blood pressure; WC, waist circumference; BMI, body mass index; BF, body fat; FPG, fasting plasma glucose; HbA1c, glycated hemoglobin; HDL, high-density lipoprotein; LDL, low-density lipoprotein.; ^a,b^ significantly different between T1 and T3 of Pattern 1; ^c,d^ significantly differences between T1 and T2 and 3 of Pattern 3; ^e,f^ significantly different between T1 and T2 of Pattern 3.

## Data Availability

Not applicable.

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
