# Peer review of "Dietary Patterns of Patients with Prediabetes and Type 2 Diabetes"

_metabolites, 2023, doi:10.3390/metabo13040532_

Round 1

Reviewer 1 Report

Introduction is poorly written. It does have a fine structure (all the necessary details are included) but the content should be more concise and not written as it was picked out from a textbook. 

The text through the article is quite extensive. For example, Table 1 should be excluded or it should be in Supplementary materials if there is such a  possibility. 

Table 2 should be separated into at least two tables. At least food groups section should be separate table. Also, there are no data of statistical significance in this table.

line 104 - Instead of left arm, blood pressure is measured on non-dominant arm. Some people are left-handed or have conditions which prevents the measurement on their arm (such as AV fistula). 

Except prediabetes and type 2 diabetes we do not know a lot about this patients.

Table 5 and 6 are too big. Abbreviations can help (WC, BMI, BF... include the explanations under the table). See if all the data are necessary.

Same goes for the text. T2DM could be used instead of type 2 diabetes mellitus, BP instead of blood pressure, SBP instead of systolic blood pressure and so on. Once you explain the abbreviation you continue to use it through the text. So there is no more glycated hemoglobin. After first use in the text you write HbA1c. 

The paper needs extensive revision. 

Author Response

Thank you for your constructive and positive comments which helped improve our paper.

Introduction is poorly written. It does have a fine structure (all the necessary details are included) but the content should be more concise and not written as it was picked out from a textbook. 

The text through the article is quite extensive. For example, Table 1 should be excluded or it should be in Supplementary materials if there is such a possibility. 

Response: Thank you for your constructive comments. We have revised the Introduction (p. 1-2, lines 46-80) and moved Table 1 to Supplementary materials.

Table 2 should be separated into at least two tables. At least food groups section should be separate table. Also, there are no data of statistical significance in this table.

Response: We separated Table 2 into 2 tables (Table 1 and Table 2) (p. 4-5, lines 167-173; p.5, lines 180-181), thus food groups are now presented separately in one single table (Table 2) (p.5, lines 180-181). Statistical results have been added to the tables.

line 104 - Instead of left arm, blood pressure is measured on non-dominant arm. Some people are left-handed or have conditions which prevents the measurement on their arm (such as AV fistula). 

Response: Thank you for the suggestion. We have changed left arm to the non-dominant arm (p. 3, line 122).

Except prediabetes and type 2 diabetes we do not know a lot about this patients.

Response: We’ve added the inclusion and exclusion criteria at the end of the study population section within Methods (p. 2, lines 74-86).

Table 5 and 6 are too big. Abbreviations can help (WC, BMI, BF... include the explanations under the table). See if all the data are necessary.

Same goes for the text. T2DM could be used instead of type 2 diabetes mellitus, BP instead of blood pressure, SBP instead of systolic blood pressure and so on. Once you explain the abbreviation you continue to use it through the text. So there is no more glycated hemoglobin. After first use in the text you write HbA1c. 

Response: We have reorganized both tables (p. 7, lines 220-225; p. 8, lines 233-237) and used abbreviations throughout the article.

The paper needs extensive revision. 

Response: We have revised the paper extensively taking into consideration the reviewers’ comments as well as improving the structure and presentation. As a result, the Discussion section has been significantly shortened.

Reviewer 2 Report

In the present paper Iactu and colleagues provided a cross-sectional study carried out on 587 patients suffering type 2 diabetes or prediabetes. Their statistical analyses aimed at identifying the dietary patterns of prediabetes and type 2 diabetes patients (T2D). According to their data, the authors identified three dietary patterns for prediabetic and T2D patients, respectively.

Although the topic is of interest, many issues need to be properly addressed before considering the manuscript suitable for publication.

MAJOR ISSUES

1.       Line 183: why did the authors split each pattern across three tertiles? Please motivate the rationale behind this choice within the text.

2.       Lines 43-47: a focus on the efficacy of healthy dietary patterns, such as the Mediterranean diet, as powerful tools in control and prevention of metabolic diseases could better contextualize the topic of the study (see PMID: 35889911, PMID: 34836087).

3.       The discussion paragraph is too verbose. It must be revised pointing the lights on the results obtained and on how they could impact on the body of literature.

4.       In my opinion, it is not clear what is the most relevant evidence obtained by the study. I suggest the authors an in-depth dissection of the most significative results on the basis of the objectives reported in introduction paragraph (lines 51-54). Ultimately, the question to be answered are: Do the dietary patterns identified by the authors influence the risk of T2D or prediabetes? What are the critical issues? What are the insights arising from the results that will pave the way for possible intervention?

5.       The analyses of food consumption reported in lines 233-253 are not reported in table 7, which only reports the most frequently consumed foods. Please, provide these data in a proper table. The authors could provide them as supplementary file if they will.

6.       The correlation data reposted in lines 260-275 are not shown by appropriate graph/table. Please, provide these data in a proper table. The authors could provide them as supplementary file if they will.

MINOR ISSUES

Table 5 and 6 should be re-edited. They should be horizontally oriented and reserve one page each.

Author Response

Thank you for your constructive and positive comments which helped improve our paper.

MAJOR ISSUES

  1. Line 183: why did the authors split each pattern across three tertiles? Please motivate the rationale behind this choice within the text.

Response: Based on dietary pattern scores, study participants were classified according to the tertiles categories. The three tertiles indicate adherence to the pattern. For example, patients in tertile 1 had low adherence, those in tertile 2 had medium adherence, and those in tertile 3 had high adherence to the dietary pattern. This allows evaluation of the effects of dietary adherence on biochemical and clinical parameters. We have included this explanation in the data analyses section (p. 3, lines 140-145).

  1. Lines 43-47: a focus on the efficacy of healthy dietary patterns, such as the Mediterranean diet, as powerful tools in control and prevention of metabolic diseases could better contextualize the topic of the study (see PMID: 35889911, PMID: 34836087).

Response: Thank you for directing us to these important studies. We’ve added the importance of the Mediterranean diet in control and prevention of metabolic diseases, and the corresponding references (p. 2, lines 54-58).

  1. The discussion paragraph is too verbose. It must be revised pointing the lights on the results obtained and on how they could impact on the body of literature.

Response: Thank you for your comment. We have extensively revised the Discussion section with an eye towards improving the structure, emphasize the main results clearly and discussing the main findings in the context of the current literature. As a result, the Discussion section was significantly shortened (p. 9-11, lines 279-395).

  1. In my opinion, it is not clear what is the most relevant evidence obtained by the study. I suggest the authors an in-depth dissection of the most significative results on the basis of the objectives reported in introduction paragraph (lines 51-54). Ultimately, the question to be answered are: Do the dietary patterns identified by the authors influence the risk of T2D or prediabetes? What are the critical issues? What are the insights arising from the results that will pave the way for possible intervention?

Response: We appreciate the comment. The results as well as the discussion section have been extensively revised and the main findings, including the possible interventions are now clearly presented. Also, the Introduction section has been revised accordingly to align the stated objectives with the relevant results.

  1. The analyses of food consumption reported in lines 233-253 are not reported in table 7, which only reports the most frequently consumed foods. Please, provide these data in a proper table. The authors could provide them as supplementary file if they will.

Response: We have reported this data in Table 2S which is now part of Supplementary materials.

  1. The correlation data reposted in lines 260-275 are not shown by appropriate graph/table. Please, provide these data in a proper table. The authors could provide them as supplementary file if they will.

Response: Thank you for your comment. As requested, we have presented this data in a table format which is part of Supplementary materials (Table 3S, 4S, 5S).

MINOR ISSUES

Table 5 and 6 should be re-edited. They should be horizontally oriented and reserve one page each.

Response: Thank you for the recommendation. We have made these changes and both Tables are now presented in a landscape format.

Reviewer 3 Report

The authors proposed a protocol study to explore dietary patterns in prediabetic and type 2 diabetic patients. They aimed to compare tertiles of dietary patterns in relation to age, blood pressure, anthropometric measurements, glycemic and lipid profile. In addition, they planned to compare the dietary intake of patients in the two groups. This protocol had demonstrated the relationship between dietary patterns of patients with prediabetes and type 2 diabetes. It‘s a very interesting study. The originality was confirmed. The paper was well written. The method described very well. The sample size was large. I would congrat the efforts made by the authors. But, I had a minor concern for this study before publication.

Why not use relative risk to present the impact of characteristics by the tertiles of dietary pattern scores?

Author Response

The authors proposed a protocol study to explore dietary patterns in prediabetic and type 2 diabetic patients. They aimed to compare tertiles of dietary patterns in relation to age, blood pressure, anthropometric measurements, glycemic and lipid profile. In addition, they planned to compare the dietary intake of patients in the two groups. This protocol had demonstrated the relationship between dietary patterns of patients with prediabetes and type 2 diabetes. It‘s a very interesting study. The originality was confirmed. The paper was well written. The method described very well. The sample size was large. I would congrat the efforts made by the authors. But, I had a minor concern for this study before publication.

Thank you for your constructive and positive comments which helped improve our paper.

Why not use relative risk to present the impact of characteristics by the tertiles of dietary pattern scores?

Response: Thank you for the suggestion. We have calculated the relative risk (RR) estimated as odds ratios (OR) to address the impact of tertiles on poor metabolic control of diabetes (HbA1c > 7%). This information was added in the Data analyses (p. 4, lines 152-155) as well as in the Results section (p. 9, lines 274-278 ).

Round 2

Reviewer 1 Report

line 155 - overnight fasting

line 158 - HbA1c instead glycated hemoglobin

In Table 1 it is not necessary to write categories (sex, anthropometric measurements etc.).

line 208 - It is well known what P-values indicate statistical significance so it is not necessary to point it out under the table. This applies to all tables.

There are still too many tables in the article but seems more transparent compared to the earlier draft. 

Author Response

Thank you very much for your constructive comments and suggestions.

line 155 - overnight fasting

Response:  This has been corrected (p. 3, line 133).

line 158 - HbA1c instead glycated hemoglobin

Response: Glycated hemoglobin have been replaced as requested (p. 3, line 136).

In Table 1 it is not necessary to write categories (sex, anthropometric measurements etc.).

Response: Thank you for this comment. We made these changes (p. 4, line 167-169).

line 208 - It is well known what P-values indicate statistical significance so it is not necessary to point it out under the table. This applies to all tables.

Response: We have made the appropriate changes to the tables, as requested.

There are still too many tables in the article but seems more transparent compared to the earlier draft.

Response. We have moved several table to the Supplemental Materials

Reviewer 2 Report

The authors properly addressed the issue moved. 

Author Response

Thank you for your effort in reviewing our paper.